# Spinal Manipulative Therapy for Acute Neck Pain: A Systematic Review and Meta-Analysis of Randomised Controlled Trials

**DOI:** 10.3390/jcm10215011

**Published:** 2021-10-28

**Authors:** Aleksander Chaibi, Knut Stavem, Michael Bjørn Russell

**Affiliations:** 1Head and Neck Research Group, Division for Research and Innovation, Akershus University Hospital, 1478 Oslo, Norway; m.b.russell@medisin.uio.no; 2Department for Interdisciplinary Health Sciences, Faculty of Medicine, Institute of Health and Society, University of Oslo, 0317 Oslo, Norway; 3Institute of Clinical Medicine, Akershus University Hospital, University of Oslo, 1478 Nordbyhagen, Norway; knut.stavem@medisin.uio.no; 4Department of Pulmonary Medicine, Akershus University Hospital, 1478 Lørenskog, Norway; 5Health Services Research Unit, Akershus University Hospital, 1478 Lørenskog, Norway

**Keywords:** acute neck pain, physiotherapy, chiropractic, osteopath, spinal manipulation, randomized controlled trial, systematic review, meta-analysis, appropriateness, effectiveness

## Abstract

(1) Background: Acute neck pain is common and usually managed by medication and/or manual therapy. General practitioners (GPs) hesitate to refer to manual therapy due to uncertainty about the effectiveness and adverse events (AEs); (2) Method: To review original randomized controlled trials (RCTs) assessing the effect of spinal manipulative therapy (SMT) for acute neck pain. Data extraction was done in duplicate and formulated in tables. Quality and evidence were assessed using the Cochrane Back and Neck (CBN) Risk of Bias tool and the Grading of Recommendations Assessment, Development, and Evaluation (GRADE) criteria, respectively; (3) Results: Six studies were included. The overall pooled effect size for neck pain was very large −1.37 (95% CI, −2.41, −0.34), favouring treatments with SMT compared with controls. A single study that showed that SMT was statistically significantly better than medicine (30 mg ketorolac im.) one day post-treatment, ((−2.8 (46%) (95% CI, −2.1, −3.4) vs. −1.7 (30%) (95% CI, −1.1, −2.3), respectively; *p* = 0.02)). Minor transient AEs reported included increased pain and headache, while no serious AEs were reported; (4) Conclusions: SMT alone or in combination with other modalities was effective for patients with acute neck pain. However, limited quantity and quality, pragmatic design, and high heterogeneity limit our findings.

## 1. Introduction

Acute neck pain is very common in the adult general population, as up to 50% experienced neck pain within the last year, and recurrence is frequent [1,2]. The Global Burden of Disease study ranks musculoskeletal neck pain along with low back pain as the leading cause of non-fatal disability in almost all age groups [3,4]. The point prevalence estimate of neck pain is 4.9–7.6% [5,6]. The total cost to society of neck pain is unknown; however, a recent review estimated that the annual spending on personal health care and public health for lower back and neck pain combined was USD 87.6 billion in the US alone [7].

About one third of general practitioner (GP) consultations are due to musculoskeletal pain, mainly from neck- and lower back [8]. Patients are often advised to wait for an expected favourable natural course, usually supported with analgesic medication, and/or referred to physiotherapy treatment [9,10]. Topical NSAIDs can be beneficial and muscle relaxants are recommended as a reasonable treatment choice for acute neck pain [6,11]. In people refraining from medicine or in which medicine has an insufficient effect, manual therapy has traditionally been considered as an alternative treatment option. Nevertheless, GPs refer about 8% of people with neck pain to manual therapy, which often includes spinal manipulative therapy (SMT) intervention [10]. Fear of complications associated with cervical SMT and limited support in current guidelines towards the evidence about the effectiveness are important barriers to referrals by GPs [10]. 

Randomized controlled trials (RCTs) including a placebo group provide the best approach to assessing efficacy and safety. However, most manual therapy RCTs are pragmatic or use no intervention as a control group [12,13]. A recent Cochrane review of manual therapy for acute, subacute and chronic neck pain included 51 trials (2920 participants) and reported some evidence of an effect of SMT on neck pain [14]. Another recent systematic review on the efficacy of manual therapy and exercise for treating neck pain, including 23 RCTs, reported evidence for cervical spine manipulation and exercise in favour of thoracic spine manipulation and exercise for acute- and subacute neck pain [15].

The primary objective of this systematic review of RCTs was to determine the effect of SMT on acute neck pain of less than 6 weeks in duration. Secondary objectives were to determine the pooled effect size using unimodal SMT intervention vs. control/placebo and multimodal interventions vs. control/placebo, descriptively present the effect of SMT on disability, quality of life measures and descriptively report adverse effects (AEs).

## 2. Materials and Methods

This systematic review identified RCTs that measured the effectiveness of SMT for patients with acute neck pain. It follows the preferred reporting of items for systematic reviews and meta-analyses (PRISMA) [16].

### 2.1. Data Sources and Searches

A comprehensive literature search was conducted on the medical databases Ovid MEDLINE, Embase, Cochrane Central Register of Controlled Trials (CENTRAL), CINAHL, Web of Science and OpenGrey. In order to specify and limit the search to the area of interest, the following key words were used in various combinations: “neck pain”, “neck ache”, “cervical pain”, “cervicalgia”, “cervicodynia”, “chiropractic”, “physiotherapy”, “physical therapy”, “manual therapy”, “manipulation”, “osteopathic”, ”randomized controlled trial”, and/or “controlled clinical trial”. We identified studies by a comprehensive computerized search from inception to 27 August 2020. We further restricted the search to RCTs and studies conducted in adult humans and published in English or Scandinavian. An expert librarian at the Division for Research and Innovation, Akershus University Hospital, performed the searches, while AC reviewed titles, abstracts, and full texts. In addition, reference lists of the selected RCTs and previous systematic reviews and meta-analyses were cross-checked to identify additional relevant studies.

### 2.2. Study Selection

Acute neck pain was defined as neck pain of <6 weeks duration as the primary complaint [17]. Pilot or feasibility studies were excluded, as were studies with sub-acute (6–11 weeks duration) and/or chronic neck pain (≥12 weeks duration) that did not present individual results for the acute neck pain population. Studies in which we could not determine the duration of pain and studies that did not include pain intensity as an outcome measure were also excluded. The intervention had to include SMT alone (unimodal intervention) or in combination with any other interventions (multimodal intervention). The SMT intervention could be conducted by any type of clinician, i.e., physiotherapist, chiropractor, osteopath. Physician was defined as a passive controlled manoeuvre that uses directional high-velocity low amplitude (HVLA) thrust directed at a specific joint past the physiological range of motion without exceeding the anatomical limit [18]. Simple advice, reassurance, and encouragement to continue normal activities were not considered as multimodal interventions. Any comparison group was included.

### 2.3. Data Extraction and Quality Assessment 

Two authors (AC and MBR) independently extracted data, with discrepancies resolved through consensus. Extracted information was formatted in a table and included country/year, study population, method, intervention, and results. 

The methodological quality and internal validity of the evidence were assessed by the same two authors (AC and MBR) using the Cochrane Back and Neck (CBN) Risk of Bias tool [19]. This tool has 12 items in the following domains: randomization, concealment, baseline differences, blinding (patient, care provider and outcome assessor), co-interventions, compliance, outcomes adequately addressed, drop-outs, timing, and intention-to-treat. Prior research has shown the ability of the CBN Risk of Bias tool to identify studies at an increased risk of bias using a threshold of 5 or 6 as a summary score [19]. Thus, studies were classified as higher quality (6–12 points) or lower quality (0–5 points). In case of uncertainty regarding an item, the RCT did not receive a point and was marked with not applicable (n/a) or question mark (?). 

We attempted to contact the authors of the included papers if the information appeared unclear or was highly important in order to calculate effect size. 

### 2.4. Main Outcomes

Pain intensity measured by any visual analogue scale (VAS) or numeric rating scale (NRS) was the primary outcome measure, while disability, quality of life measures and AEs were descriptively presented as secondary outcomes. 

### 2.5. Data Synthesis and Analyses

The primary analysis focused on the standardized mean difference in pain intensity between the groups receiving either SMT alone or in combination with a multimodal intervention vs. control/placebo (sham) or no treatment. All pain intensity outcomes were converted to a 0 to 10 scale to enable pooling of the results. As the included studies had different numbers of sessions and timings of assessments which complicated the analysis. We used data from the first assessment for each study, ranging from 1 h to 3 weeks after the baseline assessment. Random effects meta-analysis was conducted using the Hartung-Knapp-Sidak-Jonkman (HKSJ) method [20,21], which is recommended for analysis with few studies [22,23,24].

We grouped RCTs and calculated the pooled effect size of SMT alone vs. control/placebo and multimodal approach vs. control/placebo and conducted a similar analysis with pooled effects for VAS vs. NRSs. An effect size of >0.2 was regarded as small, >0.5 as medium, >0.8 as large, and >1.3 as very large [25]. 

Heterogeneity of the study results was analysed using the generalized I^2^ statistic; a percentage of 25%, 50%, and 75% has been suggested to indicate low, medium, and high heterogeneity, respectively [26]. We further examined heterogeneity using meta-regression with dichotomized independent variables. We assessed the impact of a priori identified sources of sources of heterogeneity: (1) length of time from baseline to follow-up (<1 week vs. ≥1 week), (2) unimodal vs. multimodal intervention, (3) type of scale used (VAS vs. NRS), (4) publication year (below/above the median: 2005–2009 vs. 2010–2013), (5) CBN risk of bias score (below/above the median: <7 vs. ≥7). We also prepared funnel plots and used Egger’s test for funnel plot asymmetry to identify possible publication bias with *p* <0.05 suggesting asymmetry [27].

As a secondary analysis, we analysed the standardized mean difference of change in pain intensity for the studies with available data using a similar method. Because there were only three studies for this analysis, we did not prepare forest plots for subgroups, funnel plots or any further analyses.

### 2.6. Rating the Body of Evidence

The evidence in the included articles was assessed using the Grading of Recommendations Assessment, Development, and Evaluation (GRADE) criteria, which uses the domains of study design limitations, inconsistency, indirectness, and imprecision in results, and was assessed as “high”, “moderate”, “low” or “very low” by KS [28,29]. 

### 2.7. Patient and Public Involvement

Patients were not involved in the development of the research question or its outcome measures, conduct of the research, or preparation of the manuscript. The findings will, however, be disseminated to patients via social media, relevant professional associations and news media.

## 3. Results

### 3.1. Study and Subject Selection and Characteristics

Six RCTs on acute neck pain including a total of 446 participants, met the inclusion criteria of this review (Figure 1). 

The studies were conducted in Australia, Spain and USA and published from 2005 to 2013. The interventions were conducted by physiotherapists or osteopaths, except one study, which recruited through 12 private physiotherapy, chiropractic, and osteopathy clinics combined. 

Two studies explicitly evaluated SMT alone, while four studies used multimodal interventions. The patients’ mean age was 34.3 years (SD 6.1), with a mean acute neck pain duration of 20.1 days (SD 22.2) and a mean pain intensity of 5.3 (SD 1.2) on a 0–10 NRS/VAS.

### 3.2. Methodological Quality

The methodological quality score ranged from 5 to 8 points (mean 6.5, SD 1.9) out of maximum score on 12 point (Table 1). Agreement between authors was 100% at each stage. Five RCTs were considered to be of good quality [30,31,32,33,34] and one of low quality (score < 6) [35]. Performance and reporting bias was introduced in all RCTs, as none of the RCTs blinded patients or published their research protocol. Detection bias was introduced in all but one RCT, which concealed the outcomes for the assessor [33]. 

### 3.3. Pain Outcomes

Table 2 gives an overview of the six individual RCTs study population, methods, intervention and results. The results focus on mean within- and between-group change in acute neck pain intensity between baseline and after the intervention. Neck pain intensity was statistically significant between the two treatment groups in five of the six studies, favouring SMT compared to other treatments. The mean pain intensity reduction, calculated by the authors was 66% (SD 19%) at 1 day to ≤1 week follow-up, 74% (SD 17%) at >1 week to ≤4 weeks follow-up, and 86% (SD 12%) at >4 weeks follow-up (Table 2). A single study showed that SMT was better than the NSAIDs (30 mg ketorolac im.) one day post-treatment, i.e., (−2.8, SD 1.7 (46%) (95% CI, −2.1, −3.4) vs. (−1.7, SD 1.7 (30%) (95% CI, −1.1, −2.3), respectively; *p* = 0.02)) (Table 2).

When pooling the unimodal and multimodal studies in the meta-analysis, those receiving SMT had lower neck pain intensity at the time of the first assessment point compared to other treatments, i.e., standardized mean difference of −1.37 (95% CI, −2.41, −0.34) (Figure 2). A similar forest plot is shown for studies with an end point of less than 1 week and studies with end points 1–3 weeks (Appendix A). The standardized mean difference in pain score between SMT and controls was lower in studies investigating neck pain with VAS as compared to NRS, i.e., in favour of SMT (Figure 3). In the secondary analysis of change scores from baseline to after the intervention, only three studies were included. The pooled standardized mean difference was −2.76 (−4.96 to −0.60), and I^2^ was 96% (Appendix A). 

Overall, the six RCT studies included were highly heterogeneous, irrespective of modality, duration of end-point assessment and pain measurement scale used, as indicated by a I^2^ > 90% for subgroups in the pooled analyses (Figure 2 and Figure 3 and Appendix A). When exploring the sources of heterogeneity in meta-regression analysis, only type of rating scale (NRS vs. VAS) was associated with a worse outcome, unstandardized beta (95% CI) 1.82 (0.09 to 3.56), whereas multimodal vs. unimodal, time to follow-up (1–3 weeks vs. <1 week), publication year (2010–2013 vs. 2005–2009), or CBN risk of bias score (<7 vs. ≥7) was not associated with pain at the time of assessment. A funnel plot of the six RCTs indicated that publication bias was possible (Appendix A), and Egger’s test (*p* = 0.003) supported asymmetry of the funnel plot. 

The overall level of the body of evidence was rated as very low [⊕○○○], i.e., we have very little confidence in the effect estimate, and the true effect is likely to be substantially different from the estimate of effect. For details, see Appendix A. 

### 3.4. Disability, Function, and Quality of Life Outcomes

Two RCTs did not report outcomes for disability, function, or quality of life measures [30,35]. 

#### Disability, Function, and Quality of Life Outcomes in Unimodal Intervention Studies

One unimodal RCT found no statistically significant between-group differences between the cervical SMT and the cervical spinal mobilization group for the neck disability index (NDI), patient-specific functional scale or health-related quality of life, respectively, from baseline to 4 and 12 weeks follow-up (all *p* ≥ 0.30) [32]. The cervical SMT group had an improved neck disability index, patient specific functional scale and health-related quality of life by 60%, 105% and 11%, respectively, at 4 weeks follow-up, and 67%, 115% and 17%, respectively, at 12 weeks follow-up, while the cervical mobilization group improved by 53%, 81% and 9%, respectively, at 4 weeks, and 63%, 100% and 16%, respectively, at 12 weeks [32].

### 3.5. Disability, Function, and Quality of Life Outcomes in Multimodal Intervention Studies

Three multimodal RCTs presented results for disability [31,33,34]. One RCT showed within-group statistically significant reduction in Neck Pain Questionnaire (NPQ) (both *p* < 0.001), while between-group statistically significant improvement was found in favour of the experimental group, i.e., electro- and thermotherapy plus thoracic SMT at post-treatment (*p* < 0.001). The electro/thermal group improved by 14%, while the electro/thermal plus thoracic SMT group improved by 46% from baseline to post-treatment [31]. 

The second RCT showed lower NDI scores for the cervical SMT group as compared to the thoracic SMT group at all follow-up time points (all *p* ≤ 0.003). The cervical SMT group reduced NDI by 38%, 69%, and 72% from baseline to 1, 4, and 24 weeks follow-up, while the thoracic group reduced NDI by 14%, 28%, and 21% from baseline to 1, 4, and 24 weeks follow-up, respectively [33]. 

The third RCT reported a statistically significant between-group difference for NDI in favour for the cervical SMT plus thoracic SMT group at 1-week follow-up (*p* < 0.001). The cervical spinal mobilizations plus thoracic SMT group and the cervical spinal mobilization group reduced NDI by 57% and 28%, respectively, from baseline to 1 week post-treatment [34].

### 3.6. Adverse Events

Two RCTs did not report AEs [30,31], and one RCT mentioned that no AEs were recorded [34]. 

Three RCTs reported AEs. 

The RCT that also administered im. ketorolac (30 mg) recorded AEs in eight participants (28%) due to medicine, i.e., arm soreness, bad taste in mouth, dizziness, drowsiness, dyspepsia, heart palpitations, light headedness, nausea or vomiting, while one participant (3%) in the osteopathic cervical SMT group reported transient feeling of a “funny” arm (without motor weakness when assessed in the emergency department) [35]. 

The second RCT reported increased neck pain in 28 participants (32%) and 24 participants (27%) in the SMT and spinal mobilization group, respectively, headache (22 (25%) and 17 participants (19%), respectively), dizziness/vertigo (7 (8%) and 6 participants (7%), respectively), nausea (4 (5%) and 7 participants (8%), respectively), paraesthesia (8 (9%) and 5 participants (6%), respectively), and “others” defined as upper limb pain, neck stiffness, fatigue, mild lower back pain and unpleasant change in spinal posture (7 (8%) and 3 participants (3%), respectively) [32]. 

The third RCT reported minor and transient increased neck pain at post-treatment in one participant (7%) in the cervical SMT group by a physiotherapist, while eight participants (80%) reported minor and transient (<24 h) increased neck pain, fatigue, headache, and upper back pain in the thoracic SMT group [33]. 

No severe or serious AEs were reported.

## 4. Discussion 

To our knowledge, this is the first systematic review on the effectiveness of SMT treating acute neck pain. The main conclusion is that SMT alone or in combination with another modality is likely to be effective in the treatment of acute neck pain, and the RCTs reported few, mild and transient AEs. 

### 4.1. Methodological Considerations

The methodological quality of manual therapy RCTs is frequently being criticised for being too low [36]. However, manual therapy studies cannot reach what is considered the gold standard in pharmacological RCTs, because the manual therapist cannot be blinded.

The included RCTs support SMT as a non-pharmacological treatment option for acute neck pain; however, the studies are very heterogeneous and comprise small numbers of subjects. Therefore, the results should be interpreted with caution. Other methodologically challenges were the lack of patient and outcome blinding, which introduces serious methodological flaws. None of the RCTs included a sham placebo intervention arm. Although experts disagree on what is an appropriate placebo for a manual therapy clinical trials [37], we have previously shown that patient blinding is possible [13,38]. Furthermore, none of the included studies included a control group arm, i.e., not receiving any form of intervention or simply await treatment till study completion. Thus, all six RCTs were pragmatic trials that compared two active treatment arms. RCTs comparing a placebo group and a control group are advantageous to pragmatic RCTs that compare two active treatment arms to produce a true net effect [12,13]. It is also important to quantify a likely placebo response in all manual therapy RCTs. Double-blinded studies are not possible because the investigator cannot be blinded for obvious reasons [13,39].

The Bone and Joint Decade 2000–2010 Task Force on neck pain presented evidence-based guidelines for primary care clinicians to inform their assessments of neck pain [40]. For grade I, complaints of neck pain may be associated with stiffness or tenderness, but no significant neurological complaints; for grade II, neck pain interferes with daily activities, but no signs or symptoms are evident to suggest major structural pathology or significant nerve root compression; for grade III, complaints of neck pain are associated with significant neurological signs; and for grade IV, neck pain includes complaints of neck pain and/or its associated disorders, and the examining clinician detects signs or symptoms suggestive of major structural pathology. None of the included RCTs used a grading guideline when including neck pain patients, although a similar classification was proposed by the Quebec Task Force in 1995 [41].

Studies with unimodal approaches isolate (statistically) the individual effects of SMT better than multimodal approaches do, unfortunately; only two studies were unimodal [30,32]. Multimodal programs may, however, better represent actual clinical practice and therefore, be more relevant. Assessing the effect of multimodal programs is nevertheless problematic, because it is difficult to isolate the impact of a specific single intervention such as SMT. This challenge persists in meta-analysis of multimodal interventions. Multimodal RCTs also introduce a major risk of contextual biases as compared to unimodal RCTs, which was the case for four of our included RCTs [31,33,34,35].

The methodological scores indicated high quality in all but one study [35]. The treatment sessions ranged from 1 to 5 sessions (mean 3.2, SD 3.3) with a trial duration from one single day to 6 months (mean 27, SD 59 days) which leaves knowledge gaps in terms of dose-response and numbers needed to treat.

### 4.2. Discussion of Results 

The 55 previous systematic reviews of manual therapy for neck pain (Figure 1) included an unspecified broad spectrum of clinical entities, i.e., acute-, sub-acute- and chronic and combined disorders, such as spinal pain with or without whiplash and/or headache disorders and/or shoulder pain and/or radiculopathy, which complicate the evaluation to draw specific conclusions. Our review rigorously restricted RCTs to acute neck pain alone, which is important in clinical practice, because acute and chronic patients respond differently to treatment [42]. 

Two of the 55 systematic reviews we retrieved included four of the RCTs in the present study, i.e., a Cochrane review (2015) [14,30,31,32,33] and a recent meta-analysis (2019) [30,31,33,34,43]. The Cochrane review concluded that for acute and sub-acute neck pain, multiple sessions of cervical SMT were more effective than medications in reducing pain and improving function at immediate- and long-term follow-up, but produced similar changes in pain, function and quality of life when compared with multiple sessions of cervical mobilisation at immediate-, short- and intermediate-term follow-up. The recent meta-analysis (2019) [43] found an effect favouring the thoracic SMT for pain (mean difference −13.63; 95% CI: −21.79, −5.46) and disability (mean difference −9.93; 95% CI: −14.38, −5.48) as compared to thoracic or cervical mobilization for mechanical neck pain. No differences were found for thoracic SMT as compared to cervical SMT. Both reviews reported that there was an increased risk of bias due to inadequate provider and participant blinding [14,43].

A recent clinical practice guideline from the Orthopedic Section of the American Physical Therapy Association recommended thoracic SMT, a program of neck range of motion exercises, and scapula-thoracic and upper extremity strengthening to enhance program adherence (level B), and cervical SMT and/or mobilization (level C) for acute neck pain with mobility deficit [44]. However, the guideline only included one of the six RCTs in the present study in their analysis [32], which questions the literature search conducted. Nevertheless, the conclusion in the practice guideline is in accordance with the findings from a recent published Danish national clinical guideline for non-surgical treatments for acute neck pain that recommended manual therapy directed to the cervical and/or thoracic spine for acute neck pain [45]. The latter guideline, however, also concluded that they had very little confidence in the effect estimate, and that the true effect was likely to be substantially different from the estimate reported [45]. 

NSAIDs are the most frequently prescribed medications by GPs worldwide and are widely used for patients with low back pain [46], but similarly unproven for acute neck pain [6,11]. To our knowledge, one randomized, placebo- and active-controlled, multi-country, multi-centre parallel group trial has investigated the effect of NSAIDs for acute neck pain and found no significant superiority for 400 mg ibuprofen plus 100 mg caffeine or 400 mg ibuprofen alone over placebo [11]. 

NSAIDs have frequent AEs, such as abdominal pain, diarrhea, edema, dry mouth, rash, dizziness, headache, tiredness, etc. [46]. In comparison, The World Health Organization acknowledges manual therapy to be a safe and effective treatment with few mild and transient AEs [47]. i.e., local tenderness and tiredness on the treatment day [48], while serious AEs are very rare [49]. Thus, AEs appears to be substantially less in manual therapy than in pharmacological management using NSAIDs. 

### 4.3. Limitations

This study has limitations. First, there were limited quantity and quality of original research. Secondly, the six included RCTs were very heterogeneous with regard to study design and results, which limit our findings. Therefore, the results should be interpreted with caution. Thirdly, non-English RCTs are missed. Fourthly, it is possible that reporting bias exists, as indicated by asymmetry of the funnel plot and because none of the trial protocols was published. Finally, although the methodological quality of studies published after 2005 was high, all studies failed to blind patients. This leaves unanswered questions about the true net-effect and safety. Omission of blinding the outcome assessor further introduces a possible bias to the results.

## 5. Conclusions

In spite of important methodological shortcomings, our analysis shows that SMT alone or in combination with another modality are likely to be effective in the treatment of acute neck pain and the RCTs reported few AEs. However, due to the large heterogeneity of the included RCTs, small sample sizes, lack of blinding, and unanswered placebo effects, future more robust RCTs are required for firm conclusions.

## Figures and Tables

**Figure 1 jcm-10-05011-f001:**
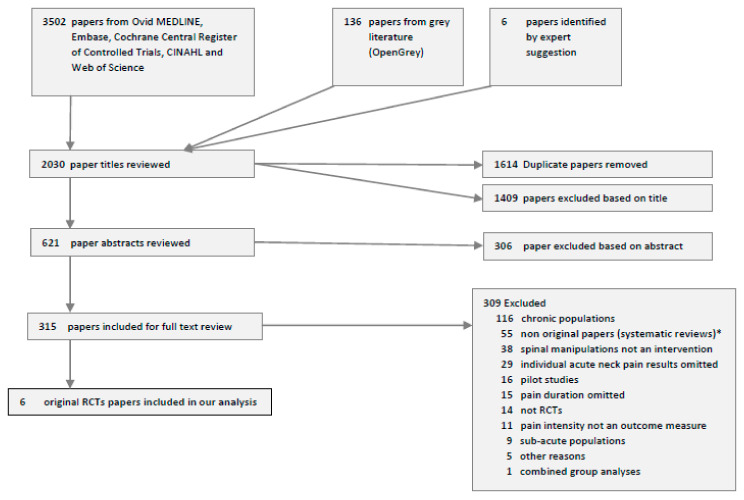
Flow chart of the literature search on acute neck pain RCT. * No original papers were retrieved from the systematic reviews.

**Figure 2 jcm-10-05011-f002:**
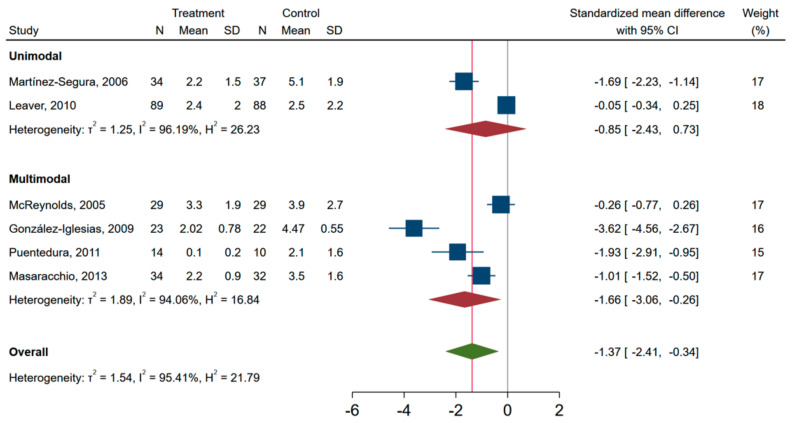
Pain intensity outcomes in randomized controlled trials of effectiveness of spinal manipulative therapy for acute neck pain with subgroups according to unimodal or multimodal interventions (*n* = 441). Between group differences at first assessment after the intervention.

**Figure 3 jcm-10-05011-f003:**
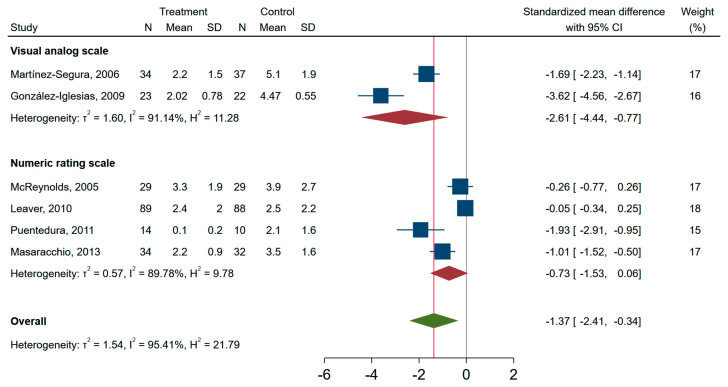
Pain intensity outcomes in randomized controlled trials of effectiveness of spinal manipulative therapy for acute neck pain with subgroups according to type rating scale used for assessment (*n* = 441). Between group differences at first assessment after the intervention.

**Table 1 jcm-10-05011-t001:** The methodological quality scores (maximum score 12 points) based on the Cochrane Back and Neck (CBN) Risk of Bias tool. Low risk of bias are scores ≥ 6, while high risk of bias are scores < 6. Yes answers = 1 point and no answers = 0 point. N/A not applicable.

	McReynolds 2005 [35]	Martínez-Segura 2006 [30]	González-Iglesias 2009 [31]	Leaver 2010 [32]	Puentedura 2011 [33]	Masaracchio 2013 [34]
1. Adequate randomization method? (selection bias)	1	1	1	1	1	1
2. Treatment allocation concealed? (selection bias)	0	0	1	1	1	1
3. Blinding of participants? (performance bias)	0	0	0	0	0	0
4. Blinding of personal? (performance bias)	0	0	0	0	0	0
5. Blinding of outcome assessor? (detection bias)	0	0	0	1	0	0
6. Incomplete outcome data adequately addressed? (attrition bias)	1	1	1	1	0	1
7. Randomized participant analysed were allocated? (attrition bias)	1	1	1	0	1	1
8. Free from selective outcome reporting? (reporting bias)	1	0	0	0	0	0
9. Similar groups at baseline?	1	1	1	1	0	1
10. Co-intervention avoided or similar?	0	1	0	1	1	0
11. Compliance with the interventions acceptable?	N/A	N/A	1	1	1	1
12. Similar timing of outcome assessment?	0	1	1	1	1	1
Total score	5	6	7	8	6	7

**Table 2 jcm-10-05011-t002:** Results from randomized clinical trials focused on pain intensity of spinal manipulative therapy.

	Country	Study Population	Methods	Intervention	Results
**Unimodal Approach**	Martíne-Segura, Spain, 2006 [30]	71 patients (26 M, 45 F) Age 20–55 yrs (mean) (Group 1; 35 ± 10 yrs Group 2; 39 ± 10 yrs) Probable acute neck pain referred from physician to a private clinic of physiotherapy and osteopathy Mean symptom duration Group 1; 4 ± 3.4 weeks Group 2; 4.5 ± 4.6 weeks Mean pain intensity Group 1; 5.7 ± 1.5 Group 2; 5.5 ± 1.5	RCT of 1-day duration Neck pain comparison at baseline and post-treatment using 10-cm line visual analogue scale (VAS 0–10)	Group 1 received a single cervical spine manipulation directed at the dysfunctional level by an osteopath (*n* = 34, 13 M, 21 F) Group 2 received a single manual mobilization procedure by an osteopath (gentle cervical side flexion and contralateral rotation which was held for 30 s) (*n* = 37, 13 M, 24 F) Drop-outs (*n* = 0)	Within-group changes showed a significant improvement in VAS at rest in group 1 and 2, (*p* < 0.001 and *p* < 0.01, respectively) Group 1 had statistically significant more reduction in VAS than group 2 at post-treatment (−3.5 (64%) (95% CI 3.1–3.9) vs. (−0.4 (7%) (95% CI 0.2–0.5), respectively; *p* < 0.001))
	Leaver, Australia, 2010 [32]	182 patients (64 M, 118 F) Age 18–70 yrs (mean) (Group 1; 38.0 ± 10.3 yrs Group 2; 39.7 ± 11.1 yrs) Acute neck pain recruited from primary care clinics in Sydney, i.e., 7 physiotherapists (*n* = 125), 5 chiropractors (*n* = 56), and 1 osteopath (*n* = 1) Mean symptom duration Group 1; 18.0 ± 19.7 days Group 2; 20.8 ± 20.4 days Mean pain intensity Group 1; 6.1 ± 2.1 Group 2; 5.9 ± 2.0	RCT of 12-weeks duration Neck pain comparison at baseline and 2- and 12-weeks using 11-point numerical rating scale (NRS 0–10) in a 2-weeks daily recording diary	Both groups received 4 treatments over a 2-week period which could include advice, reassurance, and encouragement to resume usual activities and they were refrained from seeking additional treatments Group 1 received cervical spine manipulation, i.e., high-velocity, low-amplitude thrust techniques according to their clinical judgment (*n* = 89, 37–39? M. 50–52? F) Group 2 received cervical mobilization, i.e., low-velocity, oscillating passive movement according to their clinical judgment (*n* = 88, 22–25? M, 63–66? F) Drop-outs (*n* = 5, i.e., 2 in group 1 and 3 in group 2, gender unknown)	There were no statistically significant between-group differences for NRS at 2- and 12-weeks follow-up (*p* = 0.818 and *p* = 0.504, respectively) Group 1 and 2 reduced NRS by 61% and 58%, respectively, from baseline to 2-weeks post-treatment, which sustained at 12-weeks follow-up by 74% and 76%, respectively
**Multimodal Approach**	McReynolds, USA, 2005 [35]	58 patients (26 M, 32 F) Age 18–50 yrs (mean) (Group 1; 30 ± 9 yrs Group 2; 29 ± 8 yrs) Acute neck pain examined by an osteopathic physician at three emergency teaching hospitals Median symptom duration 1 day Mean pain intensity Group 1; 5.6 ± 2.4 Group 2; 6.1 ± 1.7	RCT of 1-day duration Neck pain comparison at baseline and one-hour post-treatment using 11-point numerical rating scale (NRS 0–10)	Group 1 received 30 mg ketorolac im. by a nurse (*n* = 29, 15 M, 14 F) Group 2 received a single cervical spinal manipulation in combination with muscle energy and soft tissue techniques lasting for 5-minuttes by an osteopath physician (*n* = 29, 11 M, 18 F) Drop-outs (*n* = 0)	Both groups showed within-group statistically significant decrease in NRS at post-treatment (both *p* < 0.001) Group 2 had statistically significant more reduction in NRS than group 1 at post-treatment (−2.8 ± 1.7 (46%) (95% CI, −2.1, −3.4) vs. (−1.7 ± 1.7 (30%) (95% CI, −1.1, −2.3), respectively; *p* = 0.02)) Eighteen patients reported taking NSAIDs in the 24-hours before seeking treatment, but no statistically significant between-group difference was found at post-treatment (*p* = 0.95) Forty patients reported not taking NSAIDs in the 24-hours before seeking treatment, whereas group 2 had statistically significant more reduction in NRS than group 1 at post-treatment (−3.1 ± 1.5 (50%) (95% CI, −2.3, −3.8) vs. (−1.6 ± 1.4 (27%) (95% CI, −1.0, −2.2), respectively; *p* < 0.01))
	González-Iglesias, Spain, 2009 [31]	45 patients (24 M, 21 F) Age 18–45 yrs (mean) (Group 1; 35 ± 6 yrs Group 2; 34 ± 4 yrs) Acute neck pain referred by primary care physician to physiotherapy Mean symptom duration Group 1; 18.7 ± 3.9 days Group 2; 19.5 ± 4.5 days Mean pain intensity Group 1; 52.7 ± 5.5 Group 2; 54.7 ± 8.2	RCT of 3-weeks duration Neck pain comparison at baseline and 3-weeks post-treatment using 10-cm line visual analogue scale (VAS 0–100)	Group 1 received 5 sessions of electro- and thermotherapy over a 3-week period, i.e., infrared lamp (250 watts), located 50 cm from the patient’s neck for 15 minutes, and transcutaneous electrical nerve stimulation (100 Hz and 250 microsecond pulses) for 20 minutes using two 4 × 6 cm electrodes placed bilaterally at the spinous process of C7 vertebra, plus thoracic spine manipulation once per week for three consecutive weeks (1st-, 3rd-, and 5th visit) (*n* = 23, 12 M, 11 F) Group 2 received the same intervention as group 1 minus thoracic spine manipulation by a physiotherapist (*n* = 22, 12 M, 10 F) Drop-outs (*n* = 0)	Both groups showed within-group statistically significant reduction in VAS (both *p* < 0.001), while between-group statistically significant improvement was found in favour of group 1 at post-treatment (*p* < 0.001) Group 1 and 2 reduced VAS by 63% and 15%, respectively, at post-treatment
	Puentedura, USA, 2011 [33]	24 patients (8 M, 16 F) Age 18–60 yrs (mean) (Group 1; 34.1 ± 7.0 yrs Group 2; 33.1 ± 5.8 yrs) Acute neck pain presented to physiotherapy Mean symptom duration Group 1; 1.7 ± 7.0 days Group 2; 18.8 ± 9.3 days Mean pain intensity Group 1; 3.6 ± 1.4 Group 2; 4.6 ± 2.2	RCT of 6-months duration Neck pain comparison at baseline and 1-, 4-, and 24-weeks using 11-point numerical rating scale (NRS 0–10)	Both groups attended 5 physiotherapy sessions over a 2-week period Group 1 received 2 sessions of cervical spine manipulation, i.e., high-velocity, midrange, rotational force applied to both sides of the neck, based on pain localization and detection of joint hypomobility, plus a 2-week standardized exercise program (*n* = 10, 0–4? M, 6–10? F) Group 2 received 2 sessions of thoracic spine manipulation, i.e., 3 different manipulation techniques, mid-range high-velocity thrust, plus the same 2-week standardized exercise program (*n* = 10, 4 M, 6 F) Drop-outs (*n* = 4, i.e., 4 in group 1)	A statistically significant between-group reduction in NRS was found favouring group 1 over group 2 at 1-, and 4 weeks and 6 months (*p* = 0.003, *p* < 0.001, and *p* < 0.001, respectively) Group 1 reduced NRS by 98% at all three follow-up time points, while group 2 reduced NRS by 42%, 47%, and 36% at 1-, and 4 weeks, and 6-months follow-up, respectively
	Masaracchio, USA, 2013 [34]	66 patients (16 M, 50 F) Age 18–60 yrs (mean) (Group 1; 30.5 ± 9.5 yrs Group 2; 34.5 ± 13.3 yrs) Probable acute neck pain presented to physiotherapy or volunteered Mean symptom duration Group 1; 37.3 ± 25.3 days Group 2; 34.5 ± 26.9 days Mean pain intensity Group 1; 5.1 ± 1.2 Group 2; 4.9 ± 1.7	RCT of 1-week duration Neck pain comparison at baseline and 1-week using 11-point numerical rating scale (NRS 0–10)	Both groups attended 3 physiotherapy sessions over a 1-week period, receiving 2 treatments (session 1 and 2) and 1 assessment. Both groups were instructed in a cervical spine active ROM exercise Group 1 received the same intervention as group 2, plus 2 upper thoracic spine manipulations (T1–T3) and 2 mid thoracic spine manipulations (T4–T7) (*n* = 33, 6–7? M, 26–27? F) Group 2 received posterior-to-anterior cervical spine mobilizations (grade 2 or 3) to the spinous processes of C2–C7, each segment oscillated 10 times, followed by a 10-second rest (described by Maitland) (*n* = 31, 8–9? M, 22–23? F) Drop-outs (*n* = 2, i.e., 1 in each group)	A statistically significant between-group reduction in NRS was found favouring group 1 at 1-week follow-up (*p* < 0.001) Group 1 had 57% NRS reduction while group 2 had 29% NRS reduction at 1-week

## Data Availability

No additional data available.

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
