# Peer review of "Spinal Manipulative Therapy for Acute Neck Pain: A Systematic Review and Meta-Analysis of Randomised Controlled Trials"

_jcm, 2021, doi:10.3390/jcm10215011_

Round 1

Reviewer 1 Report

First of all, English is not my native language, so many details should be revised.

Traditionally, Manual Therapy is accused of lacking sufficiently strong scientific evidence. Many doctors do not recommend manual therapy for their patients because the evidence says it is useless. Even in cases of acute pain, they do not recommend it because it can be in danger of getting worse. It seems to me an article of great importance.

This a methodologically well-cared paper, with clinically important results.

Some minor details could be considered by the authors:

  • Line 49-50. (It is a personal doubt more than a correction) It makes reference to the use of Manual Therapy as an "alternative" when refraining from Medicine... but I personally think that Manual Therapy should be considered as first option in many cases, and not only an alternative. Maybe, we can say that "Traditionally Manual Therapy has been considered as an alternative..."
  • Lines 52-53. (Only a comment) In my clinical experience, GPs many times talk about the "lack of strong evidence" of Manual Therapy more than the risk of complications.
  • Line 59. Reference 14 is repeated
  • Search strategies could be clarified (at least annexed)
  • Languages: english and scandinavian as selection criteria could be considered a limitation. Does the authors know if other potential papers have been eliminated because of the language selection criteria?

Reviewer 2 Report

Dear authors,

this is an interesting article dealing with the effectiveness of SMT in acute neck pain. However, quality of the results is very low as stated by the authors.

Authors should discuss current therapy of acute neck pain and afterwards results of the systematic review. Furthermore, results should be discussed /compared with other articles.

Limitations should be stated in more detail highlighting the heterogeneity of the articles and the corresponding results as well as the poor quality of the screened articles.

Round 2

Reviewer 2 Report

Authors changed the manuscript according to the suggestions made by the reviewers. However, limitations of the study remain.